# ADOR: Attention Dilution and Overlap Resolver for Complex Prompts in Text-to-Image Diffusion Models

## Abstract

Text-to-image diffusion models have achieved remarkable progress, producing high-quality and realistic images. Nevertheless, these models still encounter challenges with semantic misalignment, particularly when required to understand complex prompts involving multiple objects and diverse attributes. Although several approaches have been proposed to address these issues, investigation into the causes of semantic misalignment has remained limited. In this work, we examine the behavior of cross-attention in text-to-image diffusion models and identify two key factors contributing to semantic misalignment: cross-attention overlap and cross-attention dilution. Building on these findings, we propose ADOR, a training-free framework that mitigates semantic misalignment in a single forward pass, without requiring external guidance. ADOR consists of two complementary modules: the Attention Overlap Disentangler (AO-Disentangler) and the Attention Dilution Reviver (AD-Reviver). The AO-Disentangler reduces cross-attention overlap between noun phrases via distance-based masking, thereby enhancing separation between object–attribute pairs. The AD-Reviver tackles the issue of reduced average cross-attention intensity that arises with longer prompts by applying L2-normalization or selective amplification. It ensures that semantic concepts remain represented during generation. We evaluate ADOR on standard benchmarks and demonstrate that it achieves state-of-the-art performance while preserving efficiency through its training-free, single-pass design.

## 1 Introduction

Recent advances in text-to-image diffusion models (Rombach et al., 2022; Podell et al., 2023; Peebles & Xie, 2023; Esser et al., 2024) have achieved remarkable progress in photorealistic generation from natural language. These models demonstrate exceptional capability in generating intricate details of objects and nuanced stylistic variations, enabling diverse creative and practical applications. For instance, they have been facilitating realistic scene rendering for virtual environments (Poole et al., 2022), detailed illustrations for storytelling (Liu et al., 2024), and personalized content generation (Ruiz et al., 2023).

Despite remarkable progress, existing text-to-image models struggle to accurately render complex prompts that specify multiple objects with diverse attributes (Feng et al., 2024; Meral et al., 2023; Yang et al., 2024; Wang et al., 2025). This semantic misalignment arises when the generated visual content fails to faithfully reflect the input semantics. It appears in various forms: *object entanglement*, where distinct subjects are erroneously merged into a single entity (Rassin et al., 2024; Zhuang et al., 2024); *improper attribute binding*, where characteristics such as color or texture are incorrectly assigned to objects (Li et al., 2024; Rassin et al., 2024; Zhuang et al., 2024; Meral et al., 2023); and *semantic neglect*, where entities or their specified properties are entirely omitted from the generated image (Marioriyad et al., 2025; Chefer et al., 2023; Meral et al., 2023; Rassin et al., 2024).

Previous strategies to mitigate semantic misalignment include finetuning with additional datasets (Jiang et al., 2024; Hu et al., 2024; Feng et al., 2024), optimizing latent representations during inference (Chefer et al., 2023; Li et al., 2024; Meral et al., 2023), and incorporating spatial guidance generated by large language models (Lian et al., 2024; Yang et al., 2024; Wang et al., 2025). While

effective to some extent, these methods often introduce considerable overhead, requiring additional training, dependence on external modules, or significantly increased inference times. This raises the critical challenge of how to resolve semantic misalignment effectively without external guidance or prohibitive computational cost.

To address this challenge, we propose **ADOR** (Attention Dilution and Overlap Resolver), a framework designed to mitigate the key causes of semantic misalignment in text-to-image diffusion models. ADOR is training-free, requires no external guidance, and avoids costly test-time optimization, making it both efficient and widely accessible. It comprises two complementary modules. The AO-Disentangler alleviates object entanglement and improper attribute binding by identifying attention overlap regions and applying a locality-based masking strategy, which uses unambiguous regions as anchors to ensure that only the correct object-attribute pairs contribute to the attention operation within ambiguous areas. The AD-Reviver addresses semantic neglect arising from attention dilution by rebalancing cross-attention maps, selectively amplifying the attention score of the corresponding object-attribute pair. This independence from additional training and optimization leads to markedly faster inference and improved usability compared to prior methods. Extensive experiments and ablation studies demonstrate that ADOR achieves superior performance over existing approaches while maintaining efficiency.

In summary, our contributions are as follows:

- We introduce **ADOR**, a training-free framework that mitigates semantic misalignment without requiring external guidance or test-time optimization.

- We are the first to identify and empirically validate the phenomenon of *attention dilution* in text-to-image diffusion models, establishing it as a key cause of semantic misalignment.

- We design two novel modules, the **AO-Disentangler** and **AD-Reviver**, which effectively address cross-attention overlap and cross-attention dilution, thereby removing the primary causes of semantic misalignment.

- We demonstrate that our method achieves state-of-the-art performance in T2I-CompBench (Huang et al., 2023; 2025).

## 2 RELATED WORKS

**Finetuning methods** (Feng et al., 2024; Jiang et al., 2024; Hu et al., 2024) optimize either the parameters of the pretrained model or those of auxiliary modules. ELLA (Hu et al., 2024) introduces a timestep-aware semantic connector module that bridges LLMs and pre-trained diffusion models, thereby leveraging the comprehensive language understanding capabilities of LLMs. Ranni (Feng et al., 2024) finetunes a text-to-image diffusion model on the LLMs-augmented semantic-panel dataset, which includes dense descriptions for each semantic object within an image. CoMat (Jiang et al., 2024) proposes an end-to-end finetune methodology for a text-to-image diffusion model that integrates a pretrained image-to-text model to enhance concept appearance consistency and a pretrained segmentation model to enforce proper attribute binding. While these methods enhance semantic alignment, they incur substantial computational and data costs due to the reliance on additional datasets and extensive retraining.

**Inference-time optimization methods** (Chefer et al., 2023; Li et al., 2024; Zhuang et al., 2024; Meral et al., 2023; Zhang et al., 2025; Wang et al., 2025) refine the latent feature space by applying their own task-specific loss functions during inference. Attend-and-Excite (Chefer et al., 2023) introduces a loss that constrains the maximum values of the cross-attention maps for each object to be one, ensuring that all target objects are properly attended to in the generated image. Divide & Bind (Li et al., 2024) extends this idea by incorporating the Jensen-Shannon Divergence between object attention maps and their corresponding attribute attention maps to achieve proper attribute binding. CONFORM (Meral et al., 2023) employs contrastive learning during the generation process, encouraging attention maps of matching object-attribute pairs to be closer together while pushing apart those of mismatched pairs. While these approaches sidestep the computational expense of model retraining, they suffer from the drawback of slow inference speed and substantial memory overhead.

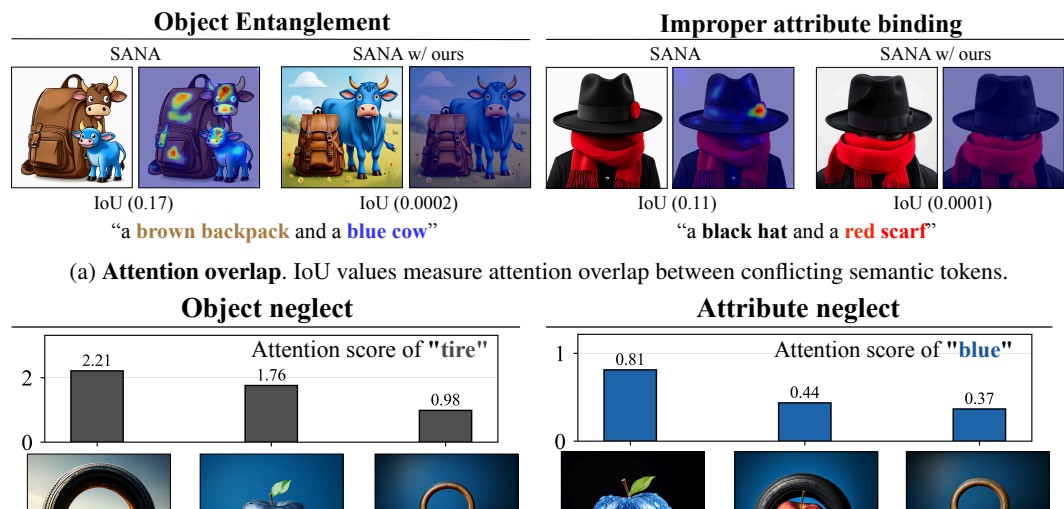

(a) **Attention overlap**. IoU values measure attention overlap between conflicting semantic tokens.

(b) **Attention dilution**. The x-axis and y-axis represent the number of concepts and attention score, respectively.

Figure 1: Analysis of semantic misalignment. (a) Comparison between base model and our method, with cross-attention heatmaps and IoU metrics highlighting *object entanglement* and *incorrect attribute binding*. (b) Example of *semantic neglect*, where key concepts are omitted; attention scores for "tire" (left) and "blue" (right) decline as prompt complexity increases.

**LLMs-based methods** (Lian et al., 2024; Yang et al., 2024; Chen et al., 2024) leverage large language models (LLMs) to extract additional conditional information from complex text prompts, thereby enabling more effective conditional image generation. LMD (Lian et al., 2024) proposes a two-stage pipeline which LLM generates an explicit layout including object locations and attributes and layout-to-image generation by controlling the diffusion model's attention maps. Self-Coherence Guidance (Wang et al., 2025) dynamically controls the cross-attention map through a mask obtained in the previous step, using ratios determined by machine learning. However, the additional priors generated by LLMs without consideration of initial noise characteristics can lead to conflicts between generated elements, resulting in degraded image quality (Ban et al., 2024; Xu et al., 2025; Dahary et al., 2025; Battash et al., 2024).

Despite the promising advances made by these prior methods, they each introduce distinct limitations, such as extensive retraining, slow inference speed, or reliance on external guidance that disregards the inherent characteristics of latent representations. To overcome these challenges, we propose a framework that effectively addresses semantic misalignment by capturing and mitigating attention conflicts at risk of semantic misalignment on the fly.

## 3 METHOD

### 3.1 OBSERVATION

**Attention overlap** Our analysis identifies two critical failure modes in cross-attention: object entanglement and improper attribute binding. To investigate these, we visualize attention heatmaps in Figure 1a, constructed by aggregating the intersections of the top 5% high-attention regions for semantically conflicting text tokens across all denoising steps and layers. For quantitative evaluation, we compute the average Intersection-over-Union (IoU) of the top 5% high-attention regions for these conflicting tokens across all denoising steps and layers. In both failure modes, we observe consis-

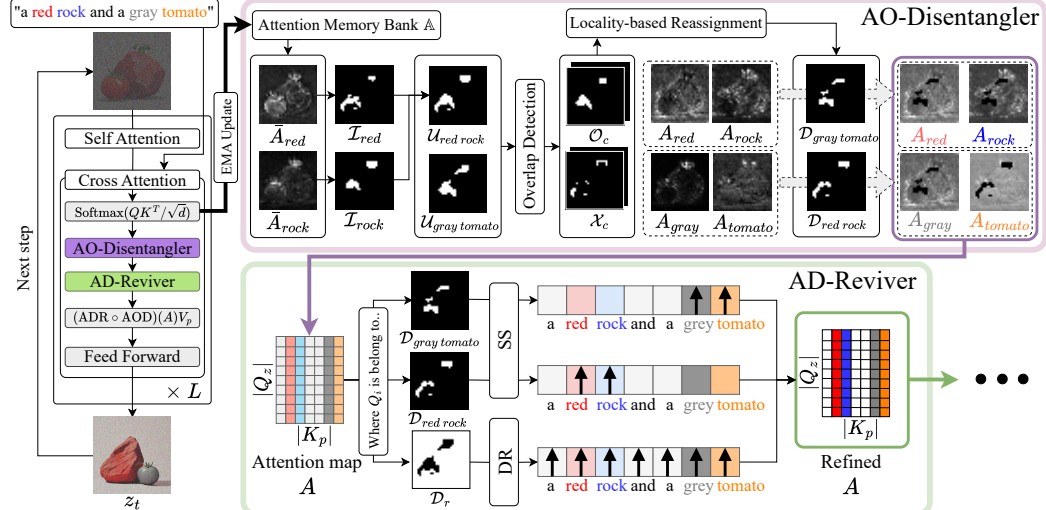

Figure 2: **Overview of the proposed ADOR framework.** ADOR modifies $L$ cross-attention layers using two components: the Attention-Overlap Disentangler (AO-Disentangler) and the Attention-Dilution Reviver (AD-Reviver). Initially, attention maps for each object and attribute token are extracted from each cross attention and do an EMA update to an attention memory bank $\mathbb{A}$. The **AO-Disentangler** then leverages $\mathbb{A}$ to detect overlapping regions between attribute-object pairs (e.g., "red rock" and "gray tomato") and performs a locality-based reassignment to create disentangled masks for each attribute-object pair. Subsequently, the **AD-Reviver** reinforces diluted attention scores for visual queries $Q_i$. For queries within the disentangled regions ($\mathcal{D}_{gray\ tomato}$ and $\mathcal{D}_{red\ rock}$), it applies selective strengthening (SS) to enhance attribute-object binding. For all remaining regions $D_r$, it performs dilution-aware rescaling (DR) to uniformly enhance attention across all text tokens, thereby mitigating semantic neglect.

tently high IoU values, indicating that a single visual token often exhibits strong correlations with unrelated text tokens. For instance, object entanglement arises when attention for distinct objects like a "brown backpack" and a "blue cow" incorrectly overlaps, while improper attribute binding occurs when attention from an attribute-object pair like "red scarf" spills onto "black hat."

**Attention dilution** We also observe a complementary issue, which we term *semantic neglect*. When multiple concepts are appended in a prompt, certain text tokens are overlooked during the denoising process, as shown in Figure 1b. This issue frequently occurs when a single visual token is forced to aggregate information from multiple semantic tokens, leading to weak or flattened correlations. The phenomenon mirrors attention dilution, a well-known problem in natural language processing, where increasing text length leads to progressively flattened attention scores (Zhang et al., 2024; Xu et al., 2024; Liu et al., 2023a). The root cause lies in the softmax operation, which enforces that attention weights sum to one. To quantify this phenomenon, we compute the sum of the top 5% of the highest attention scores for a given token within its attention map and then average across all denoising steps and cross-attention layers. As the number of key tokens increases, the attention weight multiplied by each value token tends to decrease, as illustrated in the graphs in Figure 1b.

## 3.2 OVERVIEW

As illustrated in Figure 2, our proposed method, ADOR, is a training-free, single-pass procedure that modifies cross-attention within each denoising step. The standard cross-attention mechanism, with attention map $A \in \mathbb{R}^{|Q_z| \times |K_p|}$, is defined as

$$\mathrm{CA}(Q_z, K_p, V_p) = A \cdot V_p, \quad A = \mathrm{softmax}\left(Q_z K_p^\top / \sqrt{d}\right), \tag{1}$$

where $z$ denotes the latent representation being iteratively denoised, and $p$ denotes the textual prompt that guides the generation process. The query matrix $Q_z \in \mathbb{R}^{|Q_z| \times d}$ is derived from the latent feature, while the key and value matrices, $K_p, V_p \in \mathbb{R}^{|K_p| \times d}$, are obtained from the prompt $p$.

To enhance the semantic alignment with a given text prompt $p$, we first perform syntactic parsing with an NLP library, such as spaCy (Honnibal & Montani, 2017). This syntactic information is then used to categorizes the sequence of text tokens $\mathbb{E}_p = \{p_1, ..., p_S\}$ into three groups: a set of $N$ object tokens $\mathbb{E}_o = \{o_1, ..., o_N\}$, their corresponding attribute tokens $\mathbb{E}_a = \bigcup_{i=1}^{N}\{a_1^i, ..., a_{M_i}^i\}$, where $M_i$ is the number of attributes associated with an object $o_i$; and a set of $R$ remaining tokens $\mathbb{E}_r = \{r_1, ..., r_R\}$.

ADOR modifies the attention map $A$ through the integration of AO-Disentangler (AOD) in Section 3.3 and AD-Reviver (ADR) in Section 3.4, defined as

$$\text{ADOR}(Q_z, K_p, V_p) = (\text{ADR} \circ \text{AOD})(A)V_p. \tag{2}$$

The AO-Disentangler addresses issues of *object entanglement* and *improper attribute binding*, while the AD-Reviver specifically focuses on *semantic neglect*. In the following sections, we provide a detailed description of two components, explaining how each component is designed to resolve these specific semantic failures.

### 3.3 AO-Disentangler

The AO-Disentangler is designed to resolve attention overlap that arises when multiple object-attribute concepts compete for the same visual regions. Its pipeline consists of three main stages: (i) constructing an attention memory bank to stabilize identification of attention overlap region, (ii) performing attention overlap detection to identify ambiguous and exclusive regions for each concept, and (iii) applying a locality-based reassignment strategy to ensure that the visual token is uniquely assigned to the most relevant object-attribute concept.

**Attention memory bank** To stabilize identification and minimize noise-induced overfitting, we build an attention memory bank $\mathbb{A} := \{\bar{A}_e\}$, where $e \in \mathbb{E}_o \cup \mathbb{E}_a$. The accumulated attention map $\bar{A}_e \in \mathbb{R}^{H \times W}$ is updated at every cross attention layer via an exponential moving average (EMA):

$$\bar{A}_e \leftarrow (1-\alpha)\bar{A}_e + \alpha A_e, \tag{3}$$

where $A_e \in \mathbb{R}^{H \times W}$ is the current layer's attention map for that token and $\alpha \in (0, 1]$ is the EMA rate. Here, $H$ and $W$ denote the height and width of the latent representation $z$, respectively.

**Attention overlap detection** To identify attention overlap, we define object–attribute concepts $\mathcal{C} = \{c_i\}_{i=1}^{N}$, where $c_i = \{o_i, a_1^i, ..., a_{M_i}^i\}$ and its high-correlated region $\mathcal{U}_{c_i}$ as follows:

$$\mathcal{U}_{c_i} = \mathcal{I}_{o_i} \cup \bigcup_{j=1}^{M_i} \mathcal{I}_{a_j^i}, \quad \mathcal{I}_e := \left\{ (h, w) \;\middle|\; \left[\bar{A}_e\right]_{h,w} \geq \text{Percentile}(\bar{A}_e, \beta) \right\} \tag{4}$$

where $\mathcal{I}_e$ is the set of spatial indices from the attention map $\bar{A}_e$ corresponding to a score above the $\beta$ percentile threshold. Given the object-attribute concept region $\mathcal{U}_c$, we can compute the exclusive attention region $\mathcal{X}_c$ and the attention overlap region $\mathcal{O}_c$ as follows:

$$\mathcal{X}_{c_i} = \mathcal{U}_{c_i} \setminus \bigcup_{j \neq i} \mathcal{U}_{c_j}, \quad \mathcal{O}_{c_i} = \mathcal{U}_{c_i} \setminus \mathcal{X}_{c_i}, \tag{5}$$

where $\mathcal{X}_c$ represents the unambiguous anchor regions exclusively associated with a single object-attribute concept, whereas $\mathcal{O}_c$ represents the semantically conflicted regions that must be reassigned.

**Locality-based reassignment** To resolve ambiguity in the overlap regions $\mathcal{O}_c$, we construct the disentangled region $\mathcal{D}_c$ by reassigning each ambiguous point to a single concept among the set of competing concepts based on spatial proximity:

$$\mathcal{D}_{c_i} := \mathcal{X}_{c_i} \cup \left\{ \mathbf{x} \;\middle|\; \mathbf{x} \in \mathcal{O}_{c_i} \text{ and } c_i = \operatorname*{arg\,min}_{c_k \in C(\mathbf{x})} D(\mathbf{x}, \mathcal{X}_{c_k}) \right\}, \tag{6}$$

where $C(\mathbf{x}) = \{c_k \mid \mathbf{x} \in \mathcal{O}_{c_k}\}$ represents the collection of all concepts whose ambiguous overlap regions contain the spatial point $\mathbf{x} = (h, w)$. Proximity is measured by the point-to-set distance,

$$D(\mathbf{x}, \mathcal{X}_{c_k}) = \min_{\mathbf{y} \in \mathcal{X}_{c_k}} ||\mathbf{x} - \mathbf{y}||_2, \tag{7}$$

which computes the minimum Euclidean distance from a point to any location within an exclusive region. Given $\mathcal{D}_c$, we apply a mask $M \in \mathbb{B}^{|Q_z| \times |K_p|}$ that filters attention from tokens which belong to the other disentangled regions to the attention map $A$:

$$\text{AOD}(A) = M \odot A, \quad M_{i,j} = \begin{cases} 0 & \text{if } p_j \in \mathbb{E}_o \cup \mathbb{E}_a \text{ and } \mathbf{x}_i \in \bigcup_{c \in \mathcal{C} \setminus \kappa(p_j)} \mathcal{D}_c \\ 1 & \text{otherwise} \end{cases}, \quad (8)$$

where $\mathbf{x}_i = (h_i, w_i)$ are the 2D latent coordinates for the query index $i$, where $h_i$ and $w_i$ correspond to its position in the latent feature grid. $\odot$ represents the elementwise product and the function $\kappa(p_j)$ denotes the object-attribute concept which contains the token $p_j$. By doing so, the AO-Disentangler ensures that each ambiguous visual token is uniquely assigned to the most semantically and spatially relevant object-attribute concept, effectively resolving the semantic conflict.

## 3.4 AD-REVIVER

Building on the observation discussed in Section 3.1, we propose the AD-Reviver, an adaptive attention rescaling strategy to resolve attention dilution. It consists of two complementary components: dilution-aware rescaling, which globally stabilizes attention distributions, and selective strengthening, which locally reinforces semantically relevant tokens. Formally, we define

$$\text{ADR}(A)_{i,k} = \begin{cases} \text{SS}(A)_{i,k} & \text{if } p_j \in \mathbb{E}_o \cup \mathbb{E}_a \text{ and } \mathbf{x}_i \in \mathcal{D}_{\kappa(p_j)} \\ \text{DR}(A)_{i,k} & \text{otherwise} \end{cases}, \quad (9)$$

where SS denotes selective strengthening and DR denotes dilution-aware rescaling. In this manner, AD-Reviver enhances semantically relevant regions while mitigating attention dilution across dispersed distributions, thereby ensuring that the contribution of each semantic token is not diminished.

**Dilution-aware rescaling** To mitigate attention dilution, we rescale the cross-attention map $A$ on a per-query basis. Let $A'$ denote the query-wise normalized attention map, obtained by dividing each row by $\sum_k A_{i,k}$. The rescaled map is then defined as

$$\text{DR}(A)_{i,k} = \frac{A'_{i,k}}{||A'_{i,:}||_2}, \quad (10)$$

This normalization prevents divergence of the inverse of $\ell_2$ norm, regardless of whether attention masking is applied. The vector $A'_{i,:} \in \mathbb{R}^{|K_p|}$ represents the normalized attention scores associated with the $i$-th query token. We adopt the inverse $\ell_2$ norm as the scaling factor due to its adaptive behavior. Specifically, for a concentrated attention vector, the factor remains close to one. This minimal adjustment preserves the original scores, ensuring stable generation for these well-defined concepts. In contrast, for a dispersed (diluted) vector, the factor is close to $\sqrt{|K_p|}$, substantially amplifying the scores and alleviating semantic neglect.

**Selective strengthening** For queries corresponding to the disentangled region $\mathcal{D}_c$, we selectively enhance the concept tokens associated with the most relevant concept. The rescaling is defined as

$$\text{SS}(A)_{i,k} = \begin{cases} \lambda_{i,j} A_{i,j} & \text{if } p_j \in \mathbb{E}_o \cup \mathbb{E}_a \text{ and } \mathbf{x}_i \in \mathcal{D}_{\kappa(p_j)} \\ A_{i,j} & \text{otherwise} \end{cases},$$

$$\lambda_{i,j} = \left(1 + \frac{\sum_j^{|K_t|} \text{DR}(A)_{i,j} - \sum_j^{|K_t|} A_{i,j}}{\sum_{p_k \in \kappa(p_j)} A_{i,k}}\right). \quad (11)$$

This formulation adaptively strengthens the concept tokens in proportion to their original attention scores.

## 4 EXPERIMENT

### 4.1 EXPERIMENTAL SETTINGS

**Implementation details.** Our experiments utilize Sana (Xie et al., 2024) and PixArt-$\alpha$ (Chen et al., 2023) as base models. We generate images with 20 diffusion steps for Sana (Xie et al., 2024) and

Table 1: Quantitative comparison of models on T2I-CompBench benchmark, evaluating performance on color, shape, texture, 2D/3D spatial reasoning, non-spatial, and numeracy, in addition to inference speed, and resolution. The best scores in T2I-CompBench are highlighted in **bold**. Results marked with † are from (Huang et al., 2025) and †† are from (Feng et al., 2024). All other results are measured using the official codebases.

| Model | Base | T2I-CompBench | | | | | | | Speed (sec/image) ↓ | Resolution |
|---|---|---|---|---|---|---|---|---|---|---|
| | | Color ↑ | Shape ↑ | Texture ↑ | 2D-spatial ↑ | 3D-spatial ↑ | Non-spatial ↑ | Numeracy ↑ | | |
| SD2† | none | 0.5065 | 0.4221 | 0.4922 | 0.1342 | 0.3300 | 0.3127 | 0.4582 | 2.36 | 512×512 |
| Composable† | SD2 | 0.4063 | 0.3299 | 0.3645 | 0.0800 | 0.2847 | 0.2980 | 0.4272 | 11.88 | 512×512 |
| A&E† | SD2 | 0.6400 | 0.4517 | 0.5963 | 0.1455 | 0.3222 | 0.3109 | 0.4773 | 10.77 | 512×512 |
| Ranni†† | SD2 | 0.6893 | 0.4934 | 0.6325 | 0.3167 | – | – | – | 20.24 | 768×768 |
| PixArt-$\alpha$ | none | 0.3964 | 0.4062 | 0.4696 | 0.1994 | 0.3421 | 0.3081 | 0.4971 | 2.74 | 512×512 |
| SCG | PixArt-$\alpha$ | 0.5538 | 0.4115 | 0.4633 | 0.1921 | 0.3444 | 0.3094 | 0.5021 | 8.08 | 512×512 |
| Ours | PixArt-$\alpha$ | 0.6817 | 0.5425 | 0.6339 | 0.2190 | 0.3706 | 0.3104 | 0.5451 | 8.59 | 512×512 |
| Sana | none | 0.7703 | 0.5405 | 0.6744 | 0.3794 | 0.4128 | 0.3137 | 0.6096 | 9.57 | 1024×1024 |
| Ours | Sana | **0.8240** | **0.6143** | **0.7425** | **0.3862** | **0.4180** | **0.3149** | **0.6398** | 14.71 | 1024×1024 |

50 diffusion steps for PixArt-$\alpha$ (Chen et al., 2023). In both configurations, our framework employs an exponential moving average (EMA) rate of 0.5, a percentile-$\beta$ rate of 0.05, and a classifier-free guidance scale of 4.5 (Ho & Salimans, 2022). All experiments are conducted using a single NVIDIA GeForce RTX 3090 GPU.

**Evaluation metrics.** The T2I-CompBench (Huang et al., 2023; 2025) evaluates three main categories: attribute binding (color, shape, texture), object relationships (2D-spatial, 3D-spatial, non-spatial), and numeracy. For attribute binding, the BLIP model (Li et al., 2022) is employed to assess whether attributes such as color, shape, and texture are correctly associated with their respective objects in the generated image. The evaluation of 2D/3D spatial relationships and numeracy employs the UniDet model (Zhou et al., 2022). This model detects objects to compare their positions—using bounding box coordinates for 2D relationships and depth estimation for 3D relationships—and to verify that the number of generated objects matches the prompt. Non-spatial relationships are evaluated using CLIPScore (Radford et al., 2021; Hessel et al., 2021), which measures the alignment between the generated image and the provided text prompt by calculating the cosine similarity between their feature representations.

**Baselines.** We evaluate our method against a comprehensive set of baseline models that represent different approaches to compositional generation. Our comparison includes inference-time optimization methods such as Attend-and-Excite (A&E) (Rombach et al., 2022) and Composable Diffusion (Composable) (Liu et al., 2023b). Additionally, we include Ranni (Feng et al., 2024) and self-coherence guidance (SCG) (Wang et al., 2025) for comparison with a fine-tuning and LLM-based approach, respectively.

## 4.2 COMPARISON WITH OTHER MODELS ON T2I-COMPBENCH

**Quantitative results.** As shown in Table 1, we evaluate baseline performance on T2I-CompBench (Huang et al., 2023; 2025). When applied to Sana (Xie et al., 2024), our method achieves state-of-the-art results with consistent improvements across all categories. Notably, the performance gains are most pronounced for attribute types such as color (+7.0%), shape (+13.6%), and texture (+10.1%). We attribute this significant improvement to our method's targeted mitigation of a critical vulnerability in cross-attention: its tendency to incorrectly bind or dilute semantic signals across multiple concepts in complex prompts. On 512×512 generation, our PixArt-$\alpha$ implementation demonstrates strong efficiency (8.59 sec/image) while outperforming SD2-based Composable Diffusion (11.88 sec/image) and A&E (10.77 sec/image) in both speed and accuracy. Although SCG achieves marginally faster inference (8.08 sec/image), our model achieves substantially higher accuracy, with improvements of +23.1% for color and +36.8% for texture, indicating that the modest computational overhead yields significant performance gains. Collectively, these results demonstrate robust improvements across semantic alignment benchmarks, particularly in attribute binding, while maintaining competitive computational efficiency.

**Qualitative results.** Figure 3 presents a qualitative comparison between baseline models and our proposed method. Across all prompts, the generated images reveal that semantic misalignment constitutes a persistent challenge for existing approaches. For prompts (a) and (b), the base models

(a) "a **blue backpack** and a **gold clock**" | (b) "a **rubber band** and a **wooden toy**" | (c) "a **yellow leaf** and a **green butterfly**"
(d) "a **fluffy cat** and a **yellow grape**" | (e) "a **blue balloon** and a **pink ribbon** and a **yellow daisy**"

Figure 3: Qualitative comparison with other models.

exhibit object neglect or entanglement, and even previous methods fail to resolve these issues fully. In contrast, our method successfully separates "backpack" and "clock" as well as "band" and "toy", while preserving their semantic independence. For prompts (c) and (d), the base models suffer from either improper attribute binding or attribute neglect. Alternative approaches still fall short of fully addressing these challenges. In contrast, our extensions correctly match attributes and objects across pairs, ensuring that none of these concepts are neglected and that their semantics are faithfully preserved in the generated results. For prompt (e), which includes more concepts, baseline models show compounded semantic misalignment. In contrast, our method separates objects and correctly binds attributes, representing all concepts. Collectively, these results demonstrate that our approach effectively mitigates semantic misalignment, yielding outputs that are more faithfully aligned with compositional prompts.

### 4.3 ABLATION ON AO-DISENTANGLER AND AD-REVIVER

**Quantitative results.** As summarized in Table 2, we assess the AO-Disentangler (AOD) and AD-Reviver (ADR) via selective ablations while holding all other conditions fixed. For BLIP-VQA, AOD provides noticeable improvements, which become even larger when ADR is added on top (e.g., PixArt-$\alpha$: +9.1% vs. +22.4% in shape). This trend is even stronger in PixArt-$\alpha$, highlighting the role of ADR in reducing dilution ef-

Table 2: Ablation studies on the proposed method. AOD and ADR denote AO-Disentangler and AD-Reviver, respectively. **Bold** denotes the best performance.

| Baseline | AOD | ADR | BLIP-VQA | | | Numeracy ↑ |
|---|---|---|---|---|---|---|
| | | | Color ↑ | Shape ↑ | Texture ↑ | |
| Sana | ✗ | ✗ | 0.7703 | 0.5405 | 0.6744 | 0.6096 |
| | ✓ | ✗ | 0.7843 | 0.5550 | 0.7076 | 0.6397 |
| | ✓ | ✓ | **0.8240** | **0.6143** | **0.7425** | **0.6398** |
| PixArt-$\alpha$ | ✗ | ✗ | 0.3964 | 0.4062 | 0.4696 | 0.4971 |
| | ✓ | ✗ | 0.4952 | 0.4432 | 0.5296 | 0.5289 |
| | ✓ | ✓ | **0.6817** | **0.5425** | **0.6339** | **0.5451** |

fects caused by diverse text tokens in cross-attention. In contrast, AOD demonstrates a dominant contribution to numeracy performance, with substantially larger improvements compared to ADR. This suggests that attention overlaps become more probable when the semantic number of objects increases through numerical expression (e.g., changing "one bear and one horse" to "two bears and three horses"), making AOD especially effective. Overall, ADR and AOD serve distinct but complementary roles, and their combination consistently yields the strongest performance across all categories.

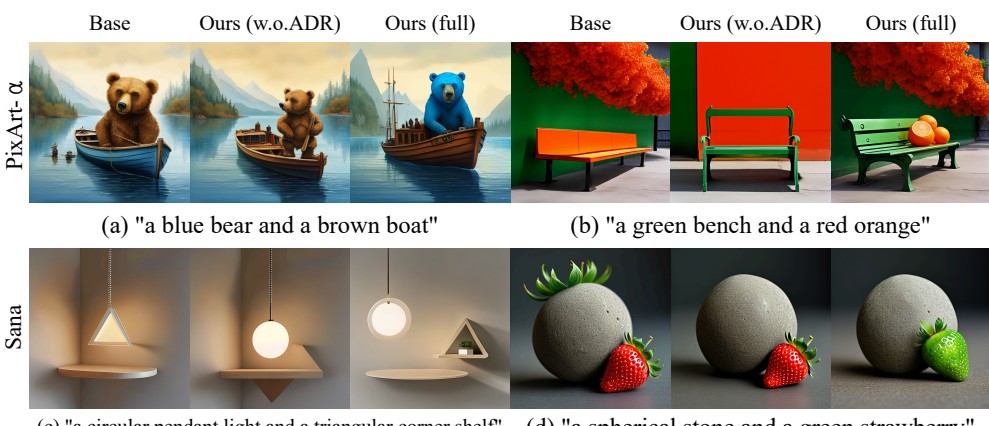

Figure 4: Ablation studies on our method.

**Qualitative results.** Figure 4 presents a qualitative ablation study highlighting the contributions of our key modules. Images generated by the base models consistently exhibit semantic misalignment. For example, in prompt (a), the attribute "blue" is incorrectly assigned to "boat" rather than "bear", while "bear" itself appears in "brown", reflecting both attribute neglect and improper binding. In prompt (b), the object "orange" is omitted, and "bench" is rendered with an incorrect color. Prompt (c) demonstrates improper attribute binding of "triangular" to "pendant light" instead of "corner shelf". Finally, in prompt (d), "stone" and "strawberry" are entangled into a single incoherent object, and "green" is insufficient from the strawberry. When the AOD module is added to the base, these issues are partially alleviated. Improper attribute binding is corrected across prompts (a), (b), and (c), while object entanglement is resolved in prompt (d). This indicates that AOD effectively disentangles mixed concepts and prevents attributes from being incorrectly matched. Nevertheless, certain deficiencies remain: "bear" and "strawberry" still lack their intended colors, and the object "orange" continues to be neglected. With the subsequent inclusion of ADR, forming our full model, these shortcomings are largely addressed. Prompts (a), (c), and (d) show attributes correctly emphasized on their respective objects. For instance, "blue" is properly bound to "bear", "triangular" to "corner shelf", and "green" to "strawberry". Moreover, ADR revives previously neglected elements, ensuring that missing objects and attributes are faithfully generated. Together, these results demonstrate that AOD and ADR complement one another, each targeting distinct sources of semantic misalignment, and that their integration is critical for achieving faithful compositional alignment.

## 5 CONCLUSION

We presented ADOR, a training-free framework that tackles the root causes of semantic misalignment in text-to-image diffusion models. Building on analysis and prior studies, we identified cross-attention overlap and cross-attention dilution as two key factors responsible for object entanglement, improper attribute binding, and neglect of visual concepts. To mitigate these issues, we designed two complementary modules. AO-Disentangler separates overlapped cross-attention signals via distance-based masking, while AD-Reviver restores balanced attention strength through normalization and selective amplification. Extensive experiments demonstrated that ADOR consistently improves semantic alignment, delivering more faithful object–attribute correspondences while preserving efficiency through a single forward pass and avoiding additional training or external guidance. These results highlight the importance of understanding and controlling cross-attention behavior as a pathway to more reliable generative modeling. Looking forward, our work opens promising avenues for attention-aware inference strategies and the extension of training-free alignment techniques to broader multimodal generation tasks, including video synthesis and text-conditioned 3D generation. By addressing the mechanisms behind semantic misalignment, ADOR offers a practical and effective step toward more semantically faithful image generation.

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

# A STATISTICAL ANALYSIS ON OBSERVATION

We extend the experiments presented in Section 3.1 with quantitative analysis.

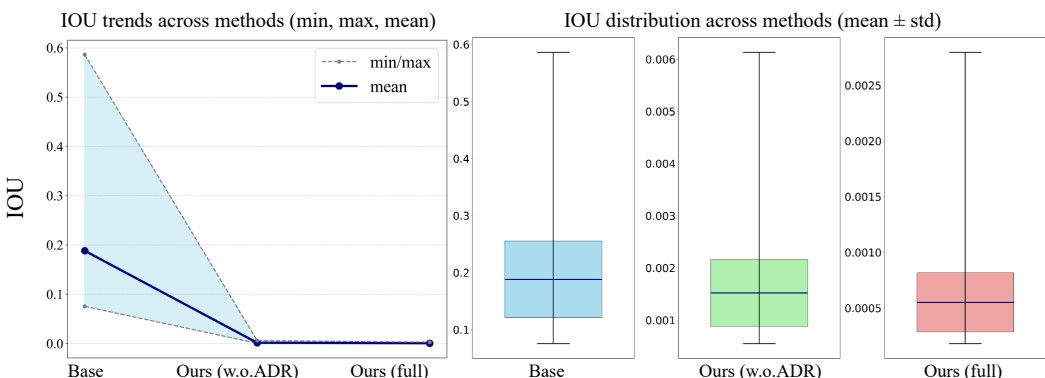

Figure 5: IOU comparison between base model and our variants. Left: a line plot summarizing IOU per method, with the solid line indicating the mean, dashed lines marking the minimum and maximum, and the shaded band spanning the min–max range. Right: per-method distributions shown as box-style, where the box reflects the mean ±1 standard deviation and whiskers denote the full range.

**Attention overlap** In Section 3.1, comparisons are restricted to the base model (Sana (Xie et al., 2024)) and our Sana variants. Here, we explicitly compare three methods: (1) the base Sana, (2) Sana without ADR (Sana + AOD), and (3) Sana with full modules (Sana + AOD + ADR). We evaluate on 800 prompts from T2I-CompBench (Huang et al., 2025) of the form "a/an ⟨attribute1⟩⟨object1⟩ and a/an ⟨attribute2⟩⟨object2⟩". Figure Figure 5 shows that adding AOD alone leads to a sharp reduction in overlap: the average IoU across the 800 prompts drops from 0.18818 to 0.00152, a 99.2% decrease. Building on this, ADR—by selectively enhancing features within the separated regions—further reduces overlap: relative to Sana + AOD, the average IoU decreases from 0.00152 to 0.00055, a 63.8% reduction. These results indicate that AOD effectively separates the attention regions corresponding to each concept, and ADR further reinforces this effect.

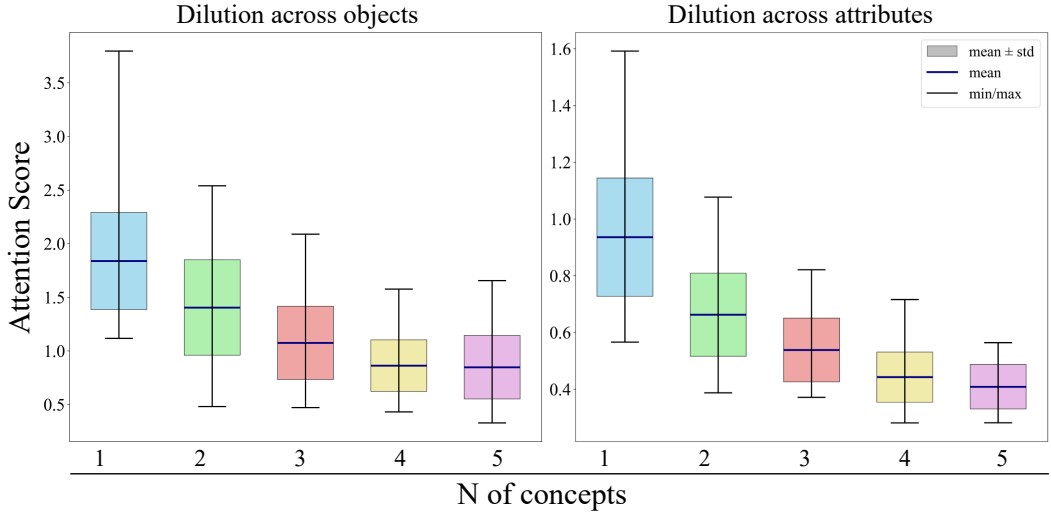

Figure 6: Attention dilution as the number of concepts increases. Left: objects; right: attributes. For each concept count (x-axis), attention scores (y-axis) are summarized with box-style plots: the box denotes mean ±1 standard deviation, the central line marks the mean, and whiskers indicate the minimum and maximum.

**Attention dilution** In Section 3.1, we examine attention dilution with up to three attribute–object pairs (i.e., up to three concepts) and a limited set of prompts. We expand this to up to five concepts and construct 100 prompts for each concept cardinality. For example, a single-concept prompt is "a/an ⟨attribute⟩⟨object⟩", and a three-concept prompt is "a/an ⟨attribute1⟩⟨object1⟩ and a/an ⟨attribute2⟩⟨object2⟩ and a/an ⟨attribute3⟩⟨object3⟩". Following this template, we manually create prompts with assistance from a large language model such as GPT (Achiam et al., 2023). For each prompt, we measure the attention scores associated with the first attribute and the first object, and summarize the mean, minimum, maximum, and standard deviation in Figure 6. For both objects and attributes, the mean attention score decreases as the number of concepts increases, following an approximately logarithmic trend. Specifically, increasing the number of concepts from 1 to 2 in attributes reduces the mean from 0.93301 to 0.66255, a 29% reduction. Overall, the results indicate that within the cross-attention mechanism of text-to-image diffusion models, increasing the number of non-padding text tokens dilutes the information allocated to each token.

## B ABLATION ON HYPERPARAMETERS

We investigate the influence of two key hyperparameters in our framework: the EMA rate and the Percentile-$\beta$ rate. As shown in Figure 7, we conduct systematic variations of each hyperparameter while holding all other conditions fixed. This analysis highlights how different choices affect performance beyond the default setting and provides practical guidance for selecting stable operating ranges.

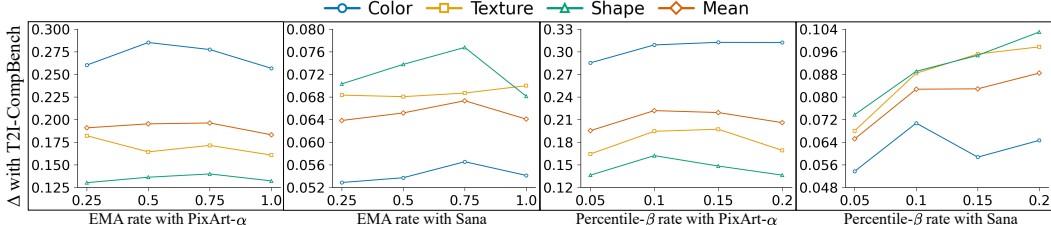

Figure 7: Effect of EMA rate and Percentile-$\beta$ rate on T2I-CompBench performance. The y-axis label indicates improvement over the corresponding base models. Results are shown for PixArt-$\alpha$ and Sana across Color, Texture, Shape, and their Mean performance.

**EMA rate.** We investigate the effect of the EMA rate while fixing the percentile-$\beta$ parameter at 0.05, varying the EMA rate over 0.25, 0.5, 0.75, 1.0. The EMA rate controls the sensitivity of the moving average: lower values dampen responsiveness to recent cross-attention signals, whereas higher values place greater weight on them. An EMA rate of 1.0 corresponds to using only the current cross-attention value (i.e., no averaging). Except for Sana in shape, both models are robust to the choice of EMA rate, showing minimal difference across categories. Overall, these findings indicate that moderate EMA rates provide the most effective balance, supporting better semantic alignment while avoiding the limitations of excessively small or overly large values.

**Percentile-$\beta$ rate.** We investigate the effect of the percentile-$\beta$ rate by fixing the EMA rate to 0.5 and varying the percentile-$\beta$ rate across 0.05, 0.1, 0.15, 0.2. The percentile-$\beta$ rate specifies a threshold for selecting indices from the EMA cross-attention weights of each text token. Based on this threshold, indices with the highest attention values are chosen in descending order. Higher percentile-$\beta$ rates result in more indices being extracted per token, while lower rates restrict the selection to fewer indices. PixArt-$\alpha$ achieves its strongest results on average at 0.1 and 0.15, with a slight decline at 0.2, while Sana continues to improve consistently except for the color category. Since larger percentile-$\beta$ rates entail extracting more indices per token and thus increase computational cost, we adopt 0.05 as the default setting.

Overall, the ablation results demonstrate that both hyperparameters yield consistent benefits and that our framework maintains stable performance without demanding fine-grained tuning. The integration of EMA and percentile-$\beta$, therefore, proves essential for mitigating semantic misalignment.

## C ADDITIONAL QUALITATIVE RESULTS

We provide additional samples below. Figure 8 demonstrates improved performance over the base model, and Figure 9 shows that our method mitigates semantic misalignment more effectively than competing models.

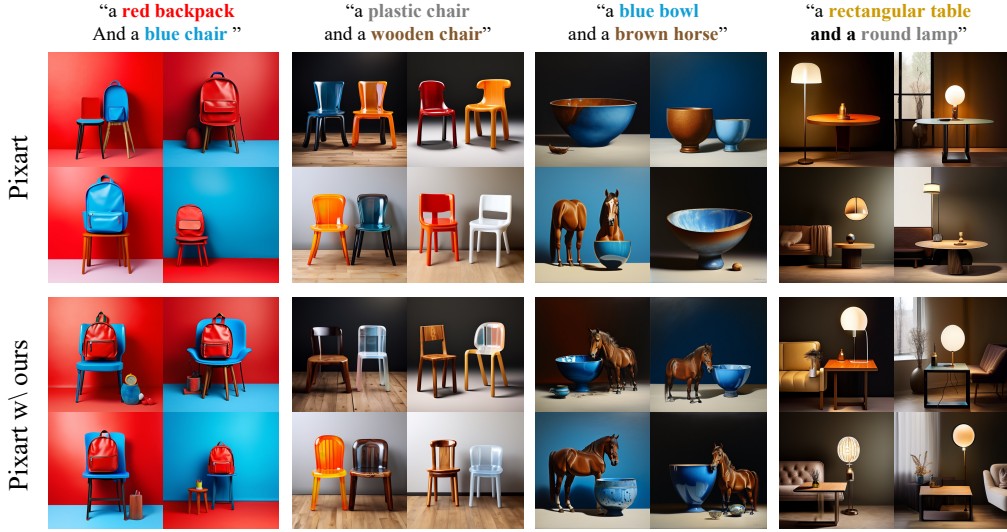

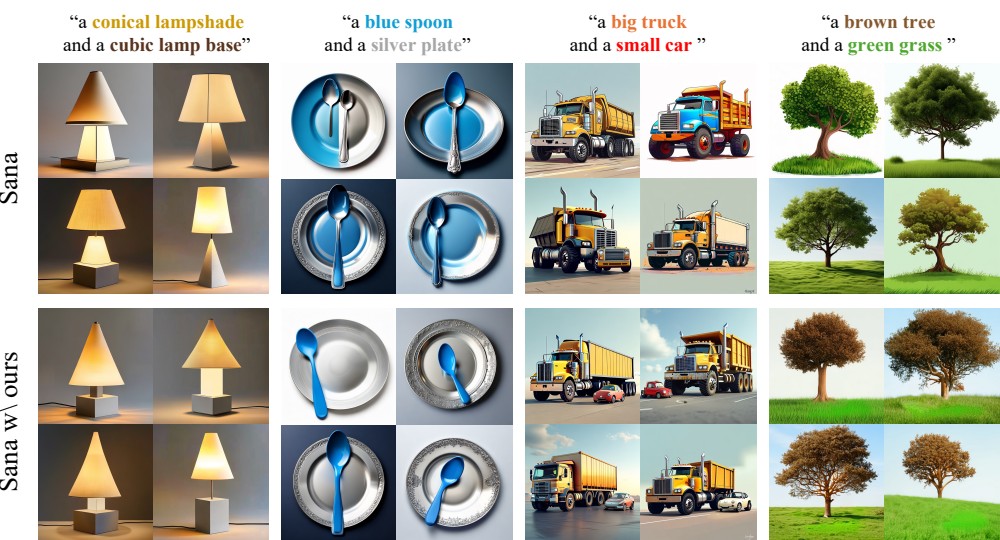

Figure 8: Additional semantically aligned images generated by our method, applied to the PixArt-$\alpha$ (top two rows) (Chen et al., 2023) and SANA (bottom two rows) (Xie et al., 2024) base models. Results are shown for two random seeds for each baseline. Zooming in is recommended for a detailed view.

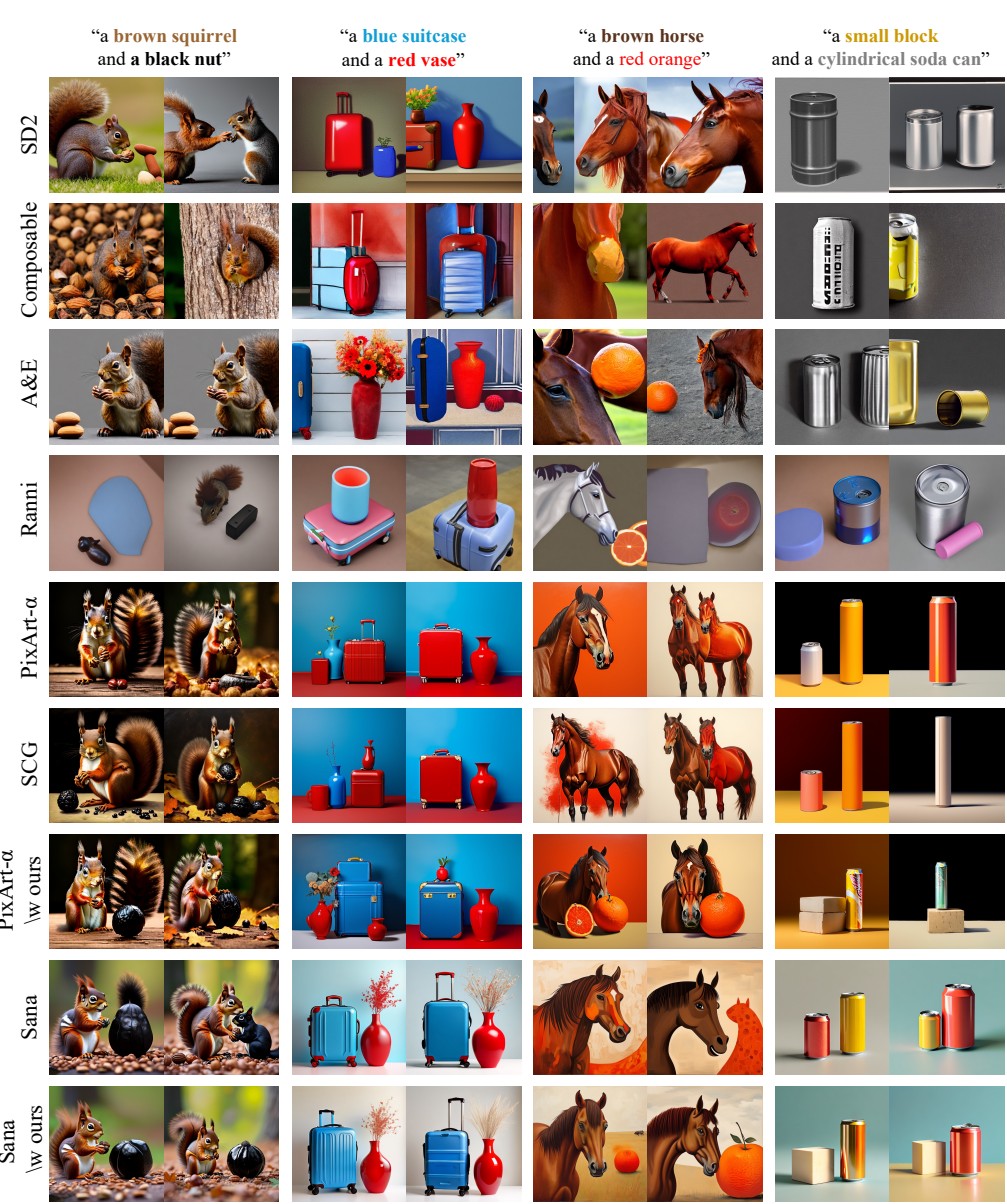

Figure 9: Additional quantitative comparison of our method against competing approaches. Results are shown for two random seeds per prompt. Zooming in is recommended for a detailed view.

## D    ETHICS STATEMENT

Following ICLR 2026 guidelines, we disclose that a Large Language Model (LLM) was utilized for assistance with grammar correction, text polishing, and the generation of prompts for additional experiments. All research contributions, experimental results, and scientific claims are entirely the work and responsibility of the authors.

