# OpenReview forum: "ADOR: Attention Dilution and Overlap Resolver for Complex Prompts in Text-to-Image Diffusion Models"
_ICLR.cc/2026/Conference — ICLR 2026 Conference Withdrawn Submission_

### Official Review · Reviewer_kZFH · 2025-10-28

**Soundness:** 3
**Presentation:** 3
**Contribution:** 3
**Rating:** 4
**Confidence:** 3

**Summary:**

This paper investigates why text-to-image diffusion models struggle with complex prompts containing multiple objects and attributes. The authors identify two primary causes for these failures: cross-attention overlap, where different concepts get entangled, and cross-attention dilution, where concepts are neglected as prompts become longer. To address this, they introduce ADOR, a training-free framework that resolves these issues during a single generation pass. By disentangling overlapping attention regions and adaptively strengthening weakened attention scores, ADOR significantly improves the model's ability to faithfully render complex scenes. The method achieves state-of-the-art performance on standard benchmarks while remaining efficient, as it requires no model retraining or external guidance.

**Strengths:**

1. The paper is very well-written and logically structured, making the core ideas easy to understand. The figures are clear and effectively illustrate the main concepts.

2. The key contribution is the insightful diagnosis of "attention overlap" and "attention dilution" as root causes of compositional failures.

**Weaknesses:**

1. The paper's core observations of attention overlap and dilution are demonstrated on weak models. It is unclear if these issues are as significant in state-of-the-art models like FLUX.1-dev or Qwen-Image. The study needs to establish that this is a general problem and not just a limitation of the specific baselines chosen.

2. The experiments successfully show improvement on a weaker baseline, Sana. However, the paper does not demonstrate that ADOR can enhance already strong compositional models. To prove its utility, the method should be tested on advanced baselines (FLUX.1-dev or Qwen-Image) to show it provides benefits beyond fixing known weaknesses in older architectures.

3. The paper claims to address "complex prompts" but limits its evaluation to T2I-CompBench, whose prompts are relatively structured. The experiments would be more convincing if they included a more challenging benchmark like DPG-Bench, which features longer and more descriptive prompts. This would better validate the method's effectiveness on truly complex text inputs.

4. Minor: The related works section could be improved by discussing recent LLM-based finetuning methods that also aim to improve prompt encoding [1].

[1] Exploring the role of large language models in prompt encoding for diffusion models.

**Questions:**

N/A

---

### Official Review · Reviewer_Vu6K · 2025-10-30

**Soundness:** 2
**Presentation:** 2
**Contribution:** 2
**Rating:** 2
**Confidence:** 3

**Summary:**

The paper presents ADOR, a training-free approach designed to improve semantic alignment in text-to-image diffusion models. It targets two key issues, overlapping attention between concepts and dilution of attention in long prompts, through two components: AO-Disentangler, which separates conflicting attention regions, and AD-Reviver, which strengthens weakened signals.

**Strengths:**

1- The method is clearly described, and the main figure effectively illustrates the overall framework.
2- The proposed modules are straightforward and can be easily integrated into existing diffusion models.

**Weaknesses:**

1- The work shows limited novelty since the ideas of disentangling overlapping attentions and enhancing weak attentions have been explored in previous works [1,2,3].
2- The experiments are limited and do not include comparisons with recent inference-time optimization approaches. The method has also not been tested on Stable Diffusion models, particularly the newer versions such as SD3 and SD3.5.
3- The evaluation metrics and benchmarks used in the paper are also limited and do not fully capture different aspects of semantic alignment and image quality.
4- The paper does not discuss the limitations of the proposed method or analyze its failure cases.


[1] Chefer, Hila, et al. "Attend-and-excite: Attention-based semantic guidance for text-to-image diffusion models."
[2] Agarwal, Aishwarya, et al. "A-star: Test-time attention segregation and retention for text-to-image synthesis."
[3] Meral, Tuna Han Salih, et al. "Conform: Contrast is all you need for high-fidelity text-to-image diffusion models."

**Questions:**

1- Could you provide a comparison between your method and recent inference-time optimization methods?
2- Could you evaluate your method on Stable Diffusion models, especially the newer versions (SD3 and SD3.5), to demonstrate its generalizability across different base models?
3- Could you extend the experiments to include additional benchmarks and more diverse evaluation metrics to provide a deeper and more balanced evaluation of your method’s effectiveness?

---

### Official Review · Reviewer_npL9 · 2025-10-31

**Soundness:** 2
**Presentation:** 2
**Contribution:** 2
**Rating:** 2
**Confidence:** 3

**Summary:**

The authors investigate why text-to-image diffusion models still struggle with multi-object or attribute-rich prompts, identifying two key causes: cross-attention overlap (where noun phrases’ attention maps bleed into one another) and attention dilution (when longer prompts reduce the per-token attention intensity). To address these, they propose a training-free method called ADOR, composed of two modules: the AO-Disentangler, which reduces overlapping attention across noun phrases via distance-based masking, and the AD-Reviver, which restores weakened attention in long prompts by either L2-normalizing or selectively amplifying attention vectors. Because ADOR works in a single forward pass without additional training or external guidance, it is efficient and practical. Empirical results on standard benchmarks show that ADOR improves semantic alignment in generated images (i.e., better matching prompt objects and attributes) and achieves state-of-the-art performance while preserving efficiency.

**Strengths:**

ADOR requires no additional model training, fine-tuning, or external guidance models. It can be directly applied to existing diffusion models like Pixalart and SANA, making it efficient, lightweight, and broadly applicable across architectures.

**Weaknesses:**

My main concern about this work lies in its limited novelty and insufficient comparison with prior baselines. The method appears to draw inspiration from several existing attention-based approaches and integrate them into a guidance-free pipeline, which functions naturally but lacks a clearly defined novel contribution. Moreover, the experiments primarily demonstrate improvements over base generation models such as PixelArt and SANA, which are not strong benchmarks. These results alone are insufficient to convincingly establish the effectiveness and generality of the proposed method.

1. No fair comparison with baselines:

The experiments lack rigorous comparisons against recent state-of-the-art compositional generation methods (e.g., A&E[1], SG[2], EBAMA[3], Astar[4]). The paper currently only compares A&E.


2. Limited evaluation scope:

The evaluation focuses mainly on qualitative visualization and compbench++, without broad quantitative or user-study validation.

3. Limited novelty:

The proposed attention disentangling and normalization ideas are relatively incremental extensions of prior attention manipulation works. For example, the IoU-based idea resembles that of A-Star, the dilution phenomenon is similar to that discussed in EBAMA, and the attention-binding concept is reminiscent of SG. Moreover, the guidance-free design is inspired by Ctrl-X, which was originally developed for controlled generation rather than text-to-image synthesis.


[1] Attend-and-excite: Attention-based semantic guidance for text-to-image diffusion models.

[2] Linguistic binding in diffusion models: Enhancing attribute correspondence through attention map alignment. NeurIPS 2023.

[3] Object-Conditioned Energy-Based Attention Map Alignment in Text-to-Image Diffusion Models. ECCV 2024.

[4] A-star: Test-time attention segregation and retention for text-to-image synthesis. ICCV 2023.

[5] Ctrl-X: Controlling Structure and Appearance for Text-To-Image Generation Without Guidance. NeurIPS 2024.

**Questions:**

1. What are the architectures of PixelArt and SANA?
Is there any differences in the attention layers used or how the proposed attention mechanism behaves across different architectural designs, such as diffusion models based on U-Net versus diffusion transformers?

2. Are there any failure cases?

3. Does the CompBench dataset include spatial attributes such as “a cat on the left of the table” or verbs like “a cat chases a dog”? In such cases, how do you define and handle the object–attribute pairs?

4. Additionally, please refer to my previously mentioned weaknesses regarding the limited novelty and insufficient comparison with prior baselines, as these remain my main concerns about the work.

---

### Official Review · Reviewer_6Xqj · 2025-11-01

**Soundness:** 3
**Presentation:** 4
**Contribution:** 2
**Rating:** 2
**Confidence:** 5

**Summary:**

ADOR is a training-free plug-in for text-to-image diffusion models that improves alignment with complex prompts by adjusting cross-attention maps during generation. It identifies two main failure sources—attention overlap (objects sharing attention regions) and attention dilution (weakened attention for long prompts)—and introduces two modules: AO-Disentangler to separate overlapping concepts and AD-Reviver to restore lost attention strength. Applied to models like Sana and PixArt-α, it boosts compositional accuracy on T2I-CompBench without retraining or external guidance.

**Strengths:**

1.	Clear and well-defined motivation – The paper identifies two fundamental causes of semantic misalignment—attention overlap and attention dilution—providing a concrete and insightful diagnosis of compositional errors in diffusion models.
2.	Training-free and efficient framework – ADOR operates in a single forward pass without retraining or external guidance, making it lightweight, efficient, and easy to integrate into existing text-to-image models.
3.	Cross-attention–based alignment strategy – The method improves semantic alignment through two complementary modules applied directly to cross-attention maps: AO-Disentangler for separating overlapping attention regions and AD-Reviver for restoring diluted attention strength.
4.	Comprehensive empirical validation – The approach shows consistent improvements across all categories of the T2I-CompBench benchmark and includes detailed ablation analyses on its components and hyperparameters.
5.	Strong clarity and interpretability – The paper is clearly written and supported by intuitive visualizations and attention heatmaps, effectively illustrating both the identified problems and the improvements achieved.

**Weaknesses:**

1.	Questionable novelty claim regarding attention dilution –
The paper’s claim of being the first to identify attention dilution is overstated. InitNO already connected subject neglect to low cross-attention activation for specific tokens, showing that weak attention causes certain objects to disappear (arXiv:2404.04650), and Marioriyad et al. (2025) discussed “insufficient attention intensity” as a factor contributing to missing entities (arXiv:2410.20972).
2.	Limited evaluation scope –
The experiments are limited to T2I-CompBench, without evaluation on other established benchmarks such as GenEval (arXiv:2310.11513), TIFA (arXiv:2303.11897), GenAI-Bench (arXiv: 2406.13743), and HRS-Bench (arXiv: 2304.05390).
3.	Incomplete literature coverage of inference-time alignment methods –
 While the paper reviews fine-tuning and attention-based inference methods, it omits several important inference-time techniques such as noise optimization and seed exploration methods, including InitNO (CVPR 2024) (arXiv:2404.04650), ReNO (2024) (arXiv:2406.04312), ImageSelect (2023) (arXiv:2305.13308), ParticleFiltering (2023) (arXiv:2312.06038), SemI (2023) (arXiv:2312.08872), RealCompo (2024) (arXiv:2402.12908), SeedRegen (2024) (arXiv:2411.18810), and CARINOX (2025) (arXiv:2509.17458). This gap weakens the positioning of ADOR relative to the broader class of inference-time scaling approaches.
4.	Unbalanced and partially unfair comparison –
The proposed method is tested on PixArt-α and Sana, but the baselines compared include models using different backbones (e.g., Stable Diffusion 2). Since architecture and resolution differences strongly affect T2I-CompBench scores, the comparison does not isolate the effect of ADOR itself.
5.	Lack of diversity in backbone evaluation –
ADOR is not validated on other diffusion architectures such as SD2, SDXL or SD3, making its scalability and adaptability uncertain, especially for larger and more commonly used models.
6.	Incomplete validation of alignment dimensions –
Although ADOR targets both attribute binding and object occurrence, most analyses and metrics focus on attributes. Object presence and count (e.g., recall of all entities in the prompt) are not quantitatively evaluated.

Minor Weaknesses

1.	No human evaluation – The study relies solely on automated benchmarks (e.g., BLIP, CLIPScore, UniDet) without conducting any human evaluation or perceptual study to assess semantic alignment quality.
2.	Limited theoretical depth – The paper explains attention dilution and overlap empirically but lacks a formal analysis of why cross-attention values flatten in long prompts.
3.	Missing failure-case discussion – Only positive qualitative examples are shown; failure cases (e.g., over-separation) are not analyzed.
4.	No isolated analysis of AD-Reviver – The rescaling/strengthening effects of AD-Reviver are not evaluated in isolation.
5.	Seed control not specified – No clear statement that random seeds are fixed consistently across all experiments/components.
6.	Missing α ablation for layer aggregation – No ablation on the α parameter used for aggregating attention across layers.

**Questions:**

Please address my mentioned concerns.

---

### Note · Authors · 2025-11-12

I have read and agree with the venue's withdrawal policy on behalf of myself and my co-authors.